# Effects of a Physical Exercise Program and Health Advice on Sedentary Behavior of Adolescents

**DOI:** 10.3390/ijerph20021064

**Published:** 2023-01-06

**Authors:** Rodolfo Carlos dos Santos Silva Filho, Jeffer Eidi Sasaki, Alex Pinheiro Gordia, Alynne Christian Ribeiro Andaki

**Affiliations:** 1Postgraduate Program in Physical Education, Federal University of Triângulo Mineiro, Uberaba 38061-500, MG, Brazil; 2Teachers Training Center, Federal University of Recôncavo da Bahia, Amargosa 45300-000, BA, Brazil

**Keywords:** adolescent, exercise, screen time, sedentary behavior

## Abstract

Sedentary behavior (SB) is a risk factor for chronic non-communicable diseases. This study aimed to assess the effects of an extracurricular physical exercise program and health advice on SB in adolescents. This was a non-randomized experimental study involving 19 adolescents divided into an intervention group (IG) and a control group (CG), aged 13–16 years from a public school in a Brazilian municipality. SB was measured using self-reports and accelerometers. The intervention included physical exercise and health advice. Repeated measures analysis of variance with a significance level of α = 5% was used. There was a time*group interaction for the subscapular fold ((pre IG = 16.30 mm vs. CG = 13.44 mm, post IG = 14.00 mm vs. CG = 15.89 mm) *p* = 0.001), and VO_2_MAX ((pre IG = 32.75 mL/kg/min vs. CG = 30.66 mL/kg/min, post IG = 35.76 mL/kg/min vs. CG = 29.28 mL/kg/min) *p* = 0.008). The accelerometer showed significant differences between groups in the total SB ((pre IG = 647.49 min/day vs. CG = 535.24 min/day, post IG = 614.02 min/day vs. CG = 586.97 min/day) *p* = 0.045), with a significant decrease in the IG. In conclusion, an extracurricular physical exercise program and health advice was effective in reducing SB in adolescents.

## 1. Introduction

Sedentary behavior (SB) is defined as any type of behavior held by an individual while awake in a sitting, lying, or reclining position which promotes an energy expenditure of ≤1.5 metabolic equivalents (METs) [1]. The constant increase in its exposure, due to the constant behavioral changes across society caused by technological advancement, has led us to adopt an increasingly sedentary lifestyle [2] and raises concerns in younger populations. Of the 7.2 billion inhabitants in the world, 42% are under 25 years old, with around 1.2 billion aged 10–19 years old. Moreover, this population is subjected to drastic personal and social changes [3], which can undermine their health.

Adolescence is the transition from childhood to adulthood [4] and is marked by the experience of new behaviors and experiences which can increase the exposure to health risk factors. This can be detrimental to adolescents’ health in the long term, since the habits acquired in this life stage may persist during adulthood [3]. In addition to the fact that this is a troubled period, we can cite the fact that SB, as the name suggests, is a “behavior”. In other words, it is something that is incorporated almost automatically in an adolescent’s life [5], which may make it even more difficult to reduce SB in this population. This generates the mobilization of important health agencies which seek to stimulate a reduction in this behavior [6,7].

There is a large amount of evidence demonstrating an association between SB and negative health outcomes in adolescence. A meta-analysis by Hermoso et al. [8] showed a positive association between suffering bullying and cyberbullying and high levels of SB. The competence to perform motor skills in adolescents appears to be another outcome impaired by exposure to SB, according to a systematic review by Santos et al. [9]. According to results of Canabrava et al. [10], SB is also considered to be an important risk factor for the development of cardiovascular diseases in this population. Research conducted with American schoolchildren showed a positive association between screen time and increased body mass index, thereby characterizing SB as an important obesity risk factor in adolescence [11]. Physical inactivity [12], high resting heart rate [13], consumption of unhealthy foods [14,15], insufficient sleep in adulthood [16], lower bone mineral density [17], and body image dissatisfaction [18] are also other negative outcomes associated with SB exposure.

These and other harms make SB an important risk factor for chronic non-communicable diseases [19,20]. Currently, they are the leading cause of death worldwide [21]. In Brazil, they are the most prevalent diseases and cause the most deaths [22,23]. Therefore, the SB studies in this population become important, especially because the prevalence of overexposure to SB in Brazilian adolescents has been shown to be quite high. According to the Brazilian Institute of Geography and Statistics [24], 60% of 9th-grade students reported watching more than two hours of television a day on weekdays. High school students in the city of *João Pessoa*, *Paraíba* showed a 79.5% prevalence of excessive screen time [25]. A review study by Silva et al. [26] showed that the majority of the analyzed studies found a prevalence of exposure to SB greater than 50%.

Despite the significant damage to health caused by SB, most experimental studies have focused on other outcomes (food and physical activity), with SB as a secondary outcome [27]. Moreover, interventions aimed at reducing SB have little effect [28,29]. Another important point concerns the methodological issues. The study of SB in adolescents is strongly based on observational research, which does not use direct methods (e.g., accelerometer) for its measurement [30]. Furthermore, the need for research evaluating the exposure to portable technologies is a gap to be covered [26]. The possession of these devices is quite high among Brazilian adolescents [31] as well as those from other nationalities [32,33]. In addition, we can note that there is a positive association between SB and the use of such equipment [34].

Therefore, it is necessary to carry out interventions that aim to reduce SB in adolescents and that use a combination of instruments of direct and indirect measures to obtain more reliable data. Accordingly, the aim of this study was to assess the effects of an extracurricular physical exercise program and health advice on SB in adolescents, as well as determining the effects of this intervention on anthropometric measures, body composition, cardiorespiratory fitness, blood pressure, and biochemical markers.

## 2. Materials and Methods

### 2.1. Study Design and Sample

This was a non-randomized experimental study. The sample was selected in a non-probabilistic way, composed of 19 adolescents from elementary and high school at a public state school in the municipality of *Uberaba*. This municipality is located in the state of Minas Gerais, and its estimated population for the year 2020 was 337,092 inhabitants [35]. The Municipal Human Development Index (MHDI) was 0.772. This index is measured on a scale of 0 to 1, with values close to 1 indicating greater human development, through the indicators of education, income, and longevity [36]. 

We included adolescents who were enrolled in classes from the 7th to the 9th grade of elementary school and the 1st grade of high school of the aforementioned school, males and females, aged 13–16 years. All of them filled out and signed the free and informed assent and consent forms together with their legal guardians. The exclusion criteria were: not using and/or answering at least one of the instruments employed to measure SB (accelerometer, the “Behavior of Adolescents from Santa Catarina” questionnaire and the “Questionnaire on Portable Technologies and Mobile Internet”), presenting any physical or mental condition that prevented him/her from participating in the proposed activities, and having a frequency below 70% in the physical exercise program. Adolescents were divided into an “intervention group”, which participated in the physical exercise program and health advice and a “control group”, which did not participate in the intervention and did not receive any type of recommendation. All adolescents were from the same school shift. The research was approved by the Committee for Ethics in Research on Human Beings of the Federal University of *Triângulo Mineiro* (protocol no. 2.915.141). This study was registered in the Brazilian Registry of Clinical Trials (register no. RBR-7xzhkn).

### 2.2. Procedures

The study was divided into three stages: “pre-intervention”, where the primary (SB) and secondary variables (socioeconomic status, anthropometry, body composition, cardiorespiratory fitness, blood pressure, and biochemical markers) were evaluated; “intervention”; and “post-intervention”, where the primary and secondary variables were retested. Data collection took place from February to July 2019.

#### 2.2.1. Socioeconomic Status

In order to evaluate economic conditions, we used a questionnaire from the Brazilian Association of Research Companies [37]. The questionnaire was filled out by the adolescents’ guardians. The socioeconomic status of the families was used to characterize the sample.

#### 2.2.2. Anthropometry

The height of each participant was measured using a personal caprice portable stadiometer (Sanny^®^, *São Bernardo do Campo*, *SP*, Brazil) with a measurement capacity ranging from 115 to 210 cm. We used the protocol designed by Lohman, Roche and Martorell [38]. Waist circumference was measured at the midpoint between the last rib and the iliac crest [39], where an anthropometric fiber measuring tape (Sanny^®^, *São Bernardo do Campo*, *SP*, Brazil) was used. Body mass was measured using a Balmak^®^ model SLIMTOP-180 digital scale (*Santa Bárbara d’ Oeste*, *SP*, Brazil), with a capacity of 180 kg and 100 g precision, following recommendations by Lohman, Roche and Martorell [38]. Measurements of the bicipital, tricipital, subscapular, and suprailiac skinfolds were obtained with a Lange skinfold caliper adipometer (Lange^®^, *London*, UK). The properly trained evaluator was the same for all groups and timepoints. The final measurement was generated from the average of three non-consecutive measurements taken on the right side of the body.

#### 2.2.3. Body Composition

Body composition was estimated using a tetrapolar electrical bioimpedance test with the Biodynamics^®^ device (King County, WA, USA), model BIA 450, and Bio Tetronic Sanny^®^ electrodes (*São Bernardo do Campo*, *SP*, Brazil). The values for lean mass, fat mass, and total body water were obtained [40]. The procedures complied with those described by Queiroga [41].

#### 2.2.4. Cardiorespiratory Fitness

Cardiorespiratory fitness was evaluated using the 12-min running/walking test proposed by Cooper [42]. The test was carried out in the sports court of the school. Queiroga’s guidelines [41] were used to perform and calculate the measurement related to the maximum oxygen volume (VO_2_max).

#### 2.2.5. Blood Pressure

Blood pressure was measured using an automatic blood pressure monitor (OMRON^®^, *São Paulo, SP*, Brazil). Three measurements were taken, the first measure was discarded, and the average of the other two was calculated according to the recommendations of the VI Brazilian Guidelines on Hypertension [43].

#### 2.2.6. Biochemical Markers

After a 12-h fast, we collected blood samples from the adolescents (7.5 mL) in vacuum tubes (VACUETTE^®^, *Americana*, Brazil). The collection was performed by a qualified professional, and conducted in a private room at the school with disposable materials. The analyzed markers were glucose, triglycerides, total cholesterol, and fractions. The reference values were glucose (70–99 mg/dL), triglycerides < 90 mg/dL, total cholesterol < 170 mg/dL, HDL-c > 45 mg/dL, and LDL-c < 110 mg/dL [44]. The samples were centrifuged at 3400 rpm for five minutes to separate the serum and plasma from other blood components. They were evaluated in a semi-automatic biochemical analyzer (Bioplus, model BIO200F, *São Paulo*, Brazil). Through the colorimetric enzymatic method, we analyzed glycemia, triglycerides, total cholesterol, and HDL-c (enzymatic-Trinder method). Moreover, we used commercial kits (BIOCLIN, Quibasa Química Básica, *Belo Horizonte*, Brazil), with manual application and the end-point method. The examinations with altered values were repeated to confirm the test.

#### 2.2.7. Sedentary Behavior

SB was evaluated using the COMPAC and Tecno-Q questionnaires and accelerometry. COMPAC [45] is comprised of 49 questions to evaluate adolescent health behaviors. We used the following questions: “How many hours a day do you watch TV?”, “How many hours a day do you use a computer and/or play video games?” and “How much time do you spend sitting and talking with friends, playing cards or dominoes, talking on the phone, driving or as a passenger, reading or studying?”. Tecno-Q [46] contains 17 questions and evaluates the possession of portable technologies and mobile internet access. In this regard, we used the following questions: “Do you have a cell phone?”, “Do you have a portable computer?”, “Do you have a tablet?”, “How much time, daily, do you spend accessing the internet through your cell phone?”, “How much time, daily, do you spend accessing the internet through your portable computer?” and “How much time, daily, do you spend accessing the internet through your tablet?”. Both questionnaires were validated for adolescents and were applied in their full versions. Outliers (reports of permanence in some behavior for 24 h a day or more) were removed from the sample.

In addition, we used ActiGraph^®^ wGT3X-BT accelerometers (Pensacola, FL, USA). The devices were fixed in the hip region on the right side of the body. The adolescents were instructed to use it for seven consecutive days, as long as possible, and to only remove them when they went to sleep at night, practice water activities, and bathe [47]. In order to validate the data, it was necessary to use at least four days (three weekdays and one weekend day), with a minimum of 10 h daily. The data were processed at a rate of 80 Hz, using epochs of 15 s [47]. The cut-off point adopted for moderate to vigorous physical activities was proposed by Evenson et al. [48]. SB was estimated as the accumulated time in ≤25 activity counts/15 s [48].

#### 2.2.8. Intervention

The intervention lasted 12 weeks and was based on the “Social Cognitive Theory” [49]. The pillars of the theory that were applied to the strategies are highlighted. The intervention included an extracurricular physical exercise program carried out at the school sports court three times a week on non-consecutive days. Each exercise session lasted 60 min. In total, 32 sessions were held. The sessions were applied by the researcher and his team. At the beginning of each session, warming-up/stretching was carried out (from five to ten minutes). In the main part of the session, recreational games and different types of ludic activities were practiced, proposed both by the research team and by the adolescents themselves, which were predominantly aerobic. At the end of the sessions, the adolescents were divided into teams and performed some competitive activities. Some of the materials used in the program were bows, balls, mattresses, cones, and ropes (intentionality). In the last session of the week, the research team reinforced the harms of high exposure to SB to the adolescents, mainly emphasizing the prevalence or incidence of chronic non-communicable diseases such as cardiovascular diseases, respiratory diseases, neoplasms, and diabetes (forecast). The research team also encouraged them to reduce screen time and time in a sitting, lying, or reclining position, mainly to hold the greatest possible number of “breaks” in this sedentary time (self-reaction).

Health advice took place as follows: a WhatsApp group was established among adolescents, their parents, or legal guardians, and the research team. Every three days, short messages were sent by a team member about the importance of reducing the SB rate and to encourage this reduction, such as “Reduce the sitting time! Your health thanks” and “Don’t get stuck! Run, play, walk!”. The messages were sent alternately in the morning and the afternoon shifts. Those who did not have a cell phone received messages via Facebook (self-reaction). The adolescents were also encouraged to make use of the “Google Fit” and “Pedometer” applications, to monitor physical activity level and steps per day and encourage their increase. Adolescents were trained to use them properly. In addition to the applications, adolescents were encouraged weekly in any of the sessions of the physical exercise program to comply with the daily recommendations for physical activity and steps (self-reaction/goal setting), to comply with the school rules of not using portable technologies during class hours, and to communicate with classmates without using such technologies. That is, to stand up and move towards this colleague, which was already allowed by the school (intentionality). In recreation time, adolescents had free access to the school sports court and its sporting materials. They were encouraged weekly in any of the sessions of the physical exercise program by the research team to use this space to move around (intentionality). Videos available on YouTube were also shown every 15 days that emphasis healthy lifestyle habits, such as “*Dicas Sesc para uma vida saudável*” and “*A saúde dos adolescentes*” (self-reflection). On the walls of the school, posters containing phrases and images about SB, its damage to health, and how to avoid the exposure to SB were fixed (self-reflection).

### 2.3. Statistical Analysis

We applied the Shapiro–Wilk test of normality. After checking the distribution of the data, we applied analysis of variance (ANOVA) of repeated measures, with the Bonferroni adjustment. Moreover, we used descriptive measures (mean and standard deviation) and the significance level adopted to be α = 5%. The data from the COMPAC and Tecno-Q questionnaires were entered and checked in EpiData 3.1 (EpiData Association, Odense, Denmark), and the Brazilian Association of Research Companies questionnaire was tabulated in the spreadsheet editor of Microsoft Office Excel. The software used during the analysis of data was the Statistical Package for the Social Sciences (SPSS), version 23.

## 3. Results

Of the 19 adolescents in the present study, 10 adolescents were assigned to the intervention group (IG), and 9 to the control group (CG). The mean frequency in the program = 70.93% (SD = 16.34). Due to the low frequency in the extracurricular program on physical exercise (31.25%), one participant in the intervention group was excluded from the research. We should mention that two adolescents in the control group did not want to use the accelerometer, but answered the two questionnaires proposed and, therefore, were not excluded from the study.

Most of the sample was composed of females (57.8%), the mean age was IG = 14.20 (SD = 1.03), and CG = 13.89 (SD = 1.26) years. The IG adolescents belonged to economic classes B1 = 27.3%, B2 = 27.3%, C1 = 18.2%, C2 = 18.2%, and D–E = 9.1%. In the CG group, the frequencies were C1 = 33.3%, C2 = 33.3%, A = 22.2%, and D–E = 11.1%. Socioeconomic classes are measured through indicators such as the ownership of electrical appliances, access to basic sanitation, and the level of education of the head of the household. Table 1 shows the distribution of the possession of portable technologies by groups in the pre- and post-intervention stages. The frequency of the adolescents’ who reported not having a cell phone in the intervention group was the same as those who reported having the device in the pre-intervention stage. In the post-intervention stage, the frequency of those who did not have a cell phone, portable computer, and tablet was higher than those who reported possession. In the control group, the cell phone possession was greater than that of those who did not have the device, both in the pre- and post-intervention stages.

Anthropometric characteristics, body composition, cardiorespiratory fitness, blood pressure, and biochemical evaluation divided by groups and the pre- and post-intervention timepoints are shown in Table 2. Among the anthropometric variables, there was a significant difference in the tricipital skinfold between the timepoints (F(1,17) = 9.950, *p* = 0.006) and time*group interaction (F(1,17) = 9.950, *p* = 0.047). There was a time*group interaction for the subscapular and suprailiac folds, (F(1,17) = 16.430, *p* = 0.001) and (F(1,17) = 9.651, *p* = 0.006), respectively. Body composition showed a significant difference between the timepoints for fat mass (F(1,17) = 4.623, *p* = 0.046), body mass (F(1,17) = 5.379, *p* = 0.033) and BMI (F(1,17) = 4.815, *p* = 0.042). The time*group interaction in cardiorespiratory fitness (VO_2_MAX) was also observed (F(1,15) = 9.346, *p* = 0.008). Only 9.4% of the biochemical results showed altered values. Total cholesterol (F(1,15) = 22.013, *p* = 0.000) and LDL-c (F(1,15) = 5.836, *p* = 0.029) showed a significant difference between the timepoints.

The mean numbers of days using the accelerometer were IG = 4.10 (SD = 0.73) and CG = 3.43 (SD = 0.53) for weekdays in the pre-intervention stage. On weekend days, the means were IG = 1.50 (SD = 0.52) and CG = 1.86 (SD = 0.37). In the post-intervention stage, the values were IG = 4.40 (SD = 0.69) and CG = 4.29 (SD = 0.75) for weekdays. On weekend days, the means were IG = 1.90 (SD = 0.31) and CG = 1.71 (SD = 0.48). The time of exposure to SB was displayed in Table 3. We noted that the weekly SB (F(1,15) = 5.549, *p* = 0.033) and the total SB (F(1,15) = 4.781, *p* = 0.045) showed a significant difference between the groups. Regarding the weekend SB, there was a time*group interaction (F(1,15) = 5.416, *p* = 0.034).

The time in a sitting position for each weekday showed a significant difference between the groups (F(1,16) = 8.193, *p* = 0.011), as well as the total mean of the time in a sitting position (F(1,16) = 7.719, *p* = 0.013), as shown in Table 4.

In Table 5, internet access time with a cell phone, portable computer, and tablet did not present a significant difference between the timepoints, groups, or time*group interaction.

## 4. Discussion

This study examined the effects of a physical exercise program and health advice on SB and anthropometric measures, body composition, cardiorespiratory fitness, blood pressure, and biochemical markers of adolescents. The major results showed that there was a statistically significant difference between the groups in weekly SB and total SB, with a reduction in the IG. For weekend SB, there was time*group interaction, with a reduction in the IG. There was a statistically significant reduction in the tricipital skinfold in the intervention group between the pre- and post-intervention stages, as well as a time*group interaction for the tricipital, subscapular, and suprailiac skinfolds and the maximum oxygen volume. Given their importance to adolescent health, these results should be emphasized.

Changes related to the skinfolds represent a significant benefit for adolescents, due to the association of these anthropometric measures with cardiovascular risk factors. Research conducted in the USA with a representative sample of children and adolescents aged 5–17 years [50] showed that high skinfold thicknesses were associated with high levels of insulin and lipids. A significant and positive association was also found between skinfolds and triglycerides for American adolescents [51]. Therefore, our results are extremely important for the health of these adolescents, thereby contributing to the prevention of cardiovascular diseases.

Cardiorespiratory fitness (VO_2_max) is an important marker of cardiovascular health in children and adolescents, in addition to being a predictor of cardiovascular profile of the later life stages [52]. It is worth highlighting the time*group interaction observed in the results, since this variable has been shown to be associated with risk factors for cardiovascular diseases [53]. A cohort study carried out with Swedish youths aged 18 years [54] showed that the number of myocardial infarctions increased with a decrease in cardiorespiratory fitness and that a 15% increase in the maximum oxygen volume was associated with a lower risk (~18%) of myocardial infarctions. In addition, improving cardiorespiratory fitness may bring other benefits to young people, such as improvements in depression symptoms, mood, anxiety, and self-esteem [53]. Accordingly, we were able to promote greater protection against the onset of cardiovascular diseases and possibly minimize other negative health outcomes.

We noted the effect of timepoint on fat mass, body mass, and body mass index, in which the post-intervention stage presented higher values than the pre-intervention stage. These changes were possibly due to skeletal growth and weight gain, which are typical of puberty, a period that is part of adolescence [55]. The absence of monitoring of the participants’ diet, coupled with the high intake of caloric foods by Brazilian adolescents [9,56] may also help to explain the increase in the values of these anthropometric variables and levels of total cholesterol and LDL-c in both groups.

With regard to SB, our primary outcome, the intervention proved to be effective in the reducing the time of exposure to SB, thereby underpinning our initial hypothesis. The accelerometers showed that the weekly SB and the total SB had a significant difference between the groups, with a reduction in the IG after the experiment. The weekend SB presented a time*group interaction, with a reduction in the IG. There was a statistically significant difference in the sitting time for each weekday and the total mean between the groups, but there was no reduction in any of the groups after the experiment. Accordingly, our results corroborate with other studies that found satisfactory results regarding reduction in exposure to SB.

School and multicomponent intervention carried out in *São Paulo* with female adolescents [57] with a primary focus on overweight and obesity showed through self-reporting that the intervention group presented a significant decrease in the use of computers on the weekend days and a decrease in the total amount of sedentary activities on weekends. Another school and multicomponent intervention with a sample of more than 1000 adolescents [58] aged 11–18 years old also showed positive changes in the amount of TV time and the use of computer and video games in the intervention group. Nevertheless, the difference in the duration of time between these interventions and ours should be emphasized. Leme et al. [57] conducted a six-month experiment and Barbosa Filho et al. [58] conducted a four-month experiment; our experiment lasted 12 weeks. Accordingly, our research contradicts the findings of a review and meta-analysis with behavioral interventions in children and adolescents [59], which reported greater effects on decreasing SB in experiments over a longer duration (six months or more). Both studies had representative samples (n = 253 and n = 1085, respectively) and used self-reporting as a tool to measure the time of exposure to SB which may underestimate the SB time at the post-intervention stage. Conversely, our intervention had a very small sample (n = 19) and we used not only self-reporting, but also an accelerometer, an objective measure of SB [60]. 

In comparison with the two aforementioned surveys, our intervention period, although it was shorter, was enough time to promote a reduction in SB. The sample size may have made it difficult to detect even greater differences between the groups in our study. In addition, different instruments may produce different results and therefore make it difficult to perform comparisons among surveys. The self-reporting technique is subject to memory bias [30] and the reporting of information may not be consistent with the participants’ reality as they may provide answers that they think are desired by the research team [61]. The accelerometer method, in turn, fails to identify the type of sedentary activity in course [62]. As an example, we can cite the intervention by Dewar et al. [63], performed with Australian girls, lasting 12 months. The applied questionnaire showed a significant reduction in the use of computers and the sum of sedentary activities for the intervention group (in relation to the control group). However, the accelerometer revealed that the two groups remained relatively stable during the studied period.

Nevertheless, some experiments have not shown significant results in reducing SB. A study carried out with 8th-grade students in public schools in the city of Tampere, Finland promoted health education classes to increase the level of physical activity and reduce the rates of SB. Accelerometry data showed that there was no difference between the intervention and control groups [64]. A study conducted by Wadolowska et al. [65] in Poland performed discussions and workshops on nutrition, diet, and physical activity, among other actions, for a period of 9-months, in order to change the lifestyle, diet, and body composition of adolescents aged 11–12. A post-intervention evaluation showed that no significant difference was found in the screen time between the groups and neither intragroup. Both interventions used different strategies to achieve their objectives and, although they did not achieve them, the use of an accelerometer [64] and the long duration of the intervention [65] were positive points of their methodologies. However, as in our study, the small sample that used the accelerometer [64] was presented as an important limitation, which perhaps made it difficult to identify differences between the groups and did not allow the generalization of results. The non-randomization of adolescents in the research groups was also a limitation [65].

An issue to be emphasized is the independence between SB and physical activity. Pablos et al. [66] performed a survey with 158 students aged 10–12 in public schools in Spain, in order to promote improvements in diet and health habits, including the reduction of SB. One of the strategies adopted was to promote the practice of physical activity, but no significant changes in SB were found between the IG and the CG. In an intervention carried out with Finnish adolescents aiming to increase the level of physical activity and reduce the rate of SB [67], equipment was provided to stimulate the practice of light physical activity in classrooms. However, no significant results in SB were noted. Such results reinforce the idea that SB and physical activity are independent variables [68] and that, consequently, they may not necessarily be inversely proportional. We support this idea, but our results have shown that the practice of physical activity may be an important ally in the quest to reduce the rate of SB.

Although its effect on the exposure to SB cannot be measured in isolation due to the multicomponent nature of the intervention, the practice of sending messages to adolescents in the intervention group via cell phone may have been an important strategy for reducing SB. The use of mobile phones and applications seems to be a means of easy accessibility and adherence to make interventions in health behaviors [69,70]. The option to include parents in the sending of messages, as a way of encouraging a reduction in the participants’ rates of SB may also have achieved the desired effect. Biddle, Petrolini, and Pearson [28] and Buchanan et al. [27] showed that family involvement in interventions brings more satisfactory results in reducing SB.

Our results corroborate the findings of systematic reviews that investigated interventions in SB. According to Biddle, Petrolini, and Pearson [28]; Young et al. [29]; and Buchanan et al. [27]; interventions focused on SB (as in the present study) showed more effective results than those that sought to intervene simultaneously in other health behaviors in conjunction with SB, such as physical activity and diet, for example. On the other hand, in the studies by Leme et al. [57] and Barbosa Filho et al. [58], it was also possible to reduce the rate of SB even though it was not the main outcome. Accordingly, it is important to investigate the best way to intervene in the SB of adolescents, experiments focused on SB or others encompassing different outcomes at the same time.

The strengths of our study should be highlighted. The measurement of SB by means of the accelerometer technique allows for more reliable data and addresses the issue of memory bias involved with the self-reporting method. The use of questionnaires provides information on the specific types of sedentary activities in which the adolescents were involved and that could not be provided by the objective measure. Moreover, it is worth highlighting the use of apps as a strategy for the reduction of SB. Although the use of these tools proves to be positive in the intervention in health behaviors, as already reported, most experiments that make use of these instruments were developed for adults, with a very low frequency of experiments for adolescents [70]. Some limitations were noted, such as the lack of surveillance for a longer period (follow-up); sample size, which does not allow for the generalization of results; and the absence of randomization of adolescents in the research groups. Another limitation may have been the evaluation of the possession of portable technologies. The employed questionnaire only considers possession of the devices owned by adolescents, i.e., the use of devices belonging to other people did not constitute possession. For future interventions in SB, we suggest that researchers follow up the results for longer periods, so that it is possible to check their maintenance.

## 5. Conclusions

We can conclude that an extracurricular physical exercise program and health advice lasting 12 weeks helped reduce the time of exposure to SB in the adolescents of this study. Moreover, we obtained important results related to the reduction of the tricipital, subscapular, and suprailiac skinfolds and in the improvement of cardiorespiratory fitness, which have huge value for the health of these adolescents as they contribute to the prevention of cardiovascular diseases.

## Figures and Tables

**Table 1 ijerph-20-01064-t001:** Distribution of portable technologies possession by groups in the pre and post-intervention stages.

Portable Technologies	IG								CG							
	Pre				Post				Pre				Post			
	YES		NOT		YES		NOT		YES		NOT		YES		NOT	
	n	%	n	%	n	%	n	%	n	%	n	%	n	%	n	%
Cell phone	5	50.0	5	50.0	3	33.3	6	66.7	6	66.7	3	33.3	6	66.7	3	33.3
Portable computer	2	20.0	8	80.0	1	10.0	9	90.0	2	22.2	7	77.8	1	12.5	7	87.5
Tablet *	2	20.0	8	80.0	1	10.0	9	90.0	1	11.1	8	88.9	-	-	8	100.0

* Omitted value (n = 1).

**Table 2 ijerph-20-01064-t002:** Anthropometric characteristics, body composition, cardiorespiratory fitness, blood pressure, and biochemical assessment by groups in the pre- and post-intervention stages.

Variables	IG			CG			*p*-Value		
	n	Mean (sd) (Pre)	Mean (sd) (Post)	n	Mean (sd) (Pre)	Mean (sd) (Post)	Time	Group	Time*Group
Height (m)	10	1.61 (0.06)	1.62 (0.06)	9	1.56 (0.11)	1.57 (0.11)	0.077	0.288	0.545
WC (cm)	10	64.70 (4.92)	63.40 (3.43)	9	65.56 (4.69)	65.00 (5.36)	0.209	0.549	0.607
Triceps SF (mm)	10	18.60 (11.40)	15.10 (9.02)	9	18.78 (5.01)	18.11 (4.51)	0.006*	0.671	0.047 *
Biceps SF (mm)	10	9.50 (5.44)	8.40 (5.64)	9	8.22 (2.90)	8.44 (3.74)	0.254	0.773	0.093
Subescapular SF (mm)	10	16.30 (7.79)	14.00 (7.33)	9	13.44 (3.64)	15.89 (4.78)	0.903	0.865	0.001 *
Suprailiac SF (mm)	10	13.90 (7.52)	11.40 (7.04)	9	13.00 (4.71)	15.22 (7.08)	0.857	0.632	0.006 *
Muscle Mass (kg)	10	43.86 (9.57)	43.92 (9.06)	9	41.23 (9.87)	41.06 (9.66)	0890	0.539	0.771
Muscle Mass %	10	85.57 (8.07)	80.39 (16.34)	9	83.43 (8.84)	81.96 (9.10)	0.093	0.954	0.334
Fat mass (kg)	10	7.49 (4.73)	8.39 (4.74)	9	8.31 (5.75)	9.06 (5.92)	0.046 *	0.759	0.855
Fat mass %	10	14.43 (8.07)	16.05 (8.55)	9	16.56 (8.84)	18.03 (9.10)	0.050	0.604	0.918
Body Mass (kg)	10	51.35 (9.70)	52.31 (8.68)	9	49.54 (10.68)	50.13 (10.18)	0.033 *	0.663	0.586
BMI (kg/m^2^)	10	19.69 (2.99)	19.99 (2.88)	9	19.66 (1.90)	19.98 (2.08)	0.042 *	0.992	0.938
BMR (cal)	10	1368.60 (298.91)	1370.20 (282.70)	9	1286.56 (307.88)	1281.22 (301.41)	0.878	0.538	0.776
VO_2_MAX (mL/kg/min)	10	32.75 (6.69)	35.76 (6.16)	7	30.66 (6.76)	29.28 (6.52)	0.276	0.191	0.008 *
SBP (mmHg)	9	106.67 (15.81)	106.67 (10.0)	9	95.56 (10.13)	103.33 (10)	0.092	0.176	0.092
DBP (mmHg)	9	63.33 (8.66)	61.11 (9.28)	9	60.00 (8.66)	61.11 (9.28)	0.715	0.679	0.282
Total cholesterol (mg/dL)	9	136.56 (20.15)	152.00 (20.19)	8	126.00 (27.09)	143.63 (20.59)	0.000 *	0.365	0.761
LDL-c (mg/dL)	9	74.66 (20.77)	79.74 (16.07)	8	52.67 (24.80)	69.33 (13.59)	0.029 *	0.067	0.218
HDL-c (mg/dL)	9	47.33 (9.67)	57.00 (13.95)	8	57.25 (14.07)	58.25 (14.58)	0.086	0.342	0.156
Triglycerides (mg/dL)	9	65.78 (15.92)	66.11 (18.41)	8	82.25 (49.64)	78.25 (31.85)	0.805	0.297	0.771
Glucose (mg/dL)	9	73.78 (6.99)	75.67 (9.04)	8	80.13 (14.89)	74.00 (11.14)	0.397	0.620	0.120

WC, waist circumference; SF, skinfold; BMI, body mass index; BMR, basal metabolic rate; VO_2_MAX, maximum oxygen uptake; SBP, systolic blood pressure; DBP, diastolic blood pressure; LDL-c, low density lipoprotein; HDL-c, high density lipoprotein; *, significant difference (*p* < 0.05).

**Table 3 ijerph-20-01064-t003:** Time of exposure to sedentary behavior (SB) obtained by the accelerometer by groups in the pre- and post-intervention stages.

Sedentary Behavior	IG			CG			*p*-Value		
	n	Mean (sd) (Pre)	Mean (sd) (Post)	n	Mean (sd) (Pre)	Mean (sd) (Post)	Time	Group	Time*Group
SB weekly (min/day)	10	656.70 (90.20)	620.26 (97.43)	7	551.72 (68.12)	577.84 (77.09)	0.859	0.033 *	0.292
SB weekend (min/day)	10	630.30 (114.69)	595.56 (158.96)	7	510.35 (73.08)	604.75 (86.26)	0.299	0.299	0.034 *
SB total (min/day)	10	647.49 (89.68)	614.02 (99.17)	7	535.24 (60.93)	586.97 (47.40)	0.709	0.045 *	0.097

SB, sedentary behavior; *, significant difference (*p* < 0.05).

**Table 4 ijerph-20-01064-t004:** Time of exposure to sedentary behavior obtained by the COMPAC questionnaire by groups in the pre- and post-intervention stages.

Variables	IG			CG			*p*-Value		
	n	Mean (sd) (Pre)	Mean (sd) (Post)	n	Mean (sd) (Pre)	Mean (sd) (Post)	Time	Group	Time*Group
Tv/day weekly (min)	10	206.90 (127.96)	154.00 (266.55)	8	217.50 (187.52)	196.25 (155.00)	0.533	0.715	0.789
Tv/day weekend (min)	8	222.13 (226.58)	158.75 (234.95)	7	130.00 (129.22)	135.71 (97.61)	0.706	0.362	0.652
Tv mean total (min)	10	198.55 (135.90)	146.28 (250.86)	9	166.98 (159.86)	178.56 (126.77)	0.717	0.995	0.570
PC-VG/day weekly (min)	9	194.44 (297.62)	33.33 (60.82)	8	26.25 (49.26)	0.00 (0.00)	0.126	0.067	0.262
PC-VG/day weekend (min)	8	229.88 (322.70)	46.25 (87.65)	8	70.00 (111.99)	15.00 (42.42)	0.093	0.129	0.347
PC-VG mean total (min)	9	197.26 (295.64)	35.55 (54.39)	9	34.44 (52.22)	3.80 (11.42)	0.094	0.058	0.242
Sitting/day weekly (min)	10	136.40 (94.67)	184.00 (246.81)	8	326.25 (276,55)	348.75 (198.95)	0.665	0.011 *	0.876
Sitting/day weekend (min)	9	152.22 (99.34)	189.44 (236.83)	6	363.33 (437.70)	400.00 (344.50)	0.657	0.115	0.997
Sitting mean total (min)	10	136.56 (85.94)	180.14 (239.80)	8	310.88 (253.32)	363.74 (210.69)	0.517	0.013 *	0.950

PC, computer; VG, video games; *, significant difference (*p* < 0.05).

**Table 5 ijerph-20-01064-t005:** Internet access time with portable technologies by groups in the pre- and post-intervention stages.

Variables	IG			CG			*p*-Value		
	n	Mean (sd) (Pre)	Mean (sd) (Post)	n	Mean (sd) (Pre)	Mean (sd) (Post)	Time	Group	Time*Group
Cell phone/day weekly (min)	6	30.00 (73.48)	0.00 (0.00)	6	200.00 (376.93)	165.00 (242.79)	0.424	0.211	0.950
Cell phone/day weekend (min)	6	30.00 (73.48)	0.00 (0.00)	6	245.00 (425.28)	240.00 (349.85)	0.518	0.181	0.642
Cell phone mean total (min)	6	30.00 (73.48)	0.00 (0.00)	6	212.85 (389.50)	186.42 (271.77)	0.420	0.200	0.959
Portable computer/day weekly (min)	10	10.00 (28.28)	0.00 (0.00)	8	22.50 (63.64)	37.50 (106.06)	0.768	0.370	0.154
Portable computer/day weekend (min)	9	2.22 (6.66)	0.00 (0.00)	8	15.00 (42.42)	33.75 (95.45)	0.367	0.326	0.256
Portable computer mean total (min)	10	7.71 (20.28)	0.00 (0.00)	8	20.35 (57.57)	36.42 (103.03)	0.608	0.349	0.156
Tablet/day weekly (min)	9	24.44 (73.33)	0.00 (0.00)	8	0.00 (0.00)	0.00 (0.00)	0.362	0.362	0.362
Tablet/day weekend (min)	9	24.44 (73.33)	0.00 (0.00)	8	0.00 (0.00)	0.00 (0.00)	0.362	0.362	0.362
Tablet/day mean total (min)	9	24.44 (73.33)	0.00 (0.00)	8	0.00 (0.00)	0.00 (0.00)	0.362	0.362	0.362

## Data Availability

The data presented in this study are available on request from the corresponding author.

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
