# Peer review of "Effects of a Physical Exercise Program and Health Advice on Sedentary Behavior of Adolescents"

_ijerph, 2023, doi:10.3390/ijerph20021064_

Round 1
Reviewer 1 Report
It is an honour for me to review the manuscript number “ ijerph-1774239 ” titled “Effects of a physical exercise program and health advice on the sedentary behavior of adolescents ” for the “IJERPH’ journal.
In this interesting article, the authors studyied the impact of extracurricular activity and health advices in order to reduce SB of adolescents from Brazil. the authors presented a non-randomized study including 19 individuals aged 13-16 years and make them practice exercice and gave health advice. The results were significant and the authors concluded that extracurricular physical exercice program and health advice was effective in reducing SB.
The paper is really good and very well written. However, there are suggestions that I must make concerning this work in order to improve the overall level of this manuscript to merit publication in this journal.
My major comments:
1) Given the presence of a few English mistakes (grammatical, typos and run-on sentence) I will advice the authors to check their manuscript another time and make a revision by native English or even a specialist.
2)Concerning references, the majority are old or even obsolete, please consider adding some new references (from 2019-2022) in your citations, they should normally represent the majority of references. Please also try to cite more papers in order to make your assertions stronger. Here are some examples of interesting papers:
*DOI: 10.3389/fendo.2020.00607
*DOI: 10.3390/nu13020374
*DOI: 10.21149/9996
*DOI: 10.1186/s12966-020-01016-4
*DOI: 10.11606/s1518-8787.2021055002917
*DOI: 10.1186/s12966-020-01037-z
3) Honestly, I am not convicted by the sample, givent the fact that these adolescents were inactive and started a physical activity, it's normal that they have results (even small results), but it will not last, they have to increase the intensity of their training in order to maintain or have better results. Thus, the conclusion is not really complete to me, because it's just temporary.
4)Can we adopt the results obtained and generalize their importance through a sample of only non-randomized sample of 19 individual? In addition we have just 10 adolescents were assigned to the intervention group which means that only10 individuals were concerned by the test.
5)Why you choose exactly the Bonferroni adjustment in your study ? can you justify the reasons for using it in you Statistical analysis part?
6) The part about health advices was not clearly explained, have you considered including a special diet like keto, OMAD, TMAD, high protein diet...etc ?
Minor comments:
1) Line 1: please remove THE
2) Line 14: Involving instead of with n=
3) Line 15: an IG and a CG
4) Line 16: using instead of by, self reports. accelerometers
5)L17: exercises
6) L21: differences
7) L43: The
8) L45: demonstrating an
9) L55: worldwide instead of in the world / diseases
10) L57: Overexposure to SB in Bra... adolescents
11) L62: significant / damages / focused
12) L67: evaluating
13) L72: to obtain
14) L80: this was
15) L95: no istead of n°
16) L160: Completed
17) L110: measured using a
18) L119: right side of the body
19) L:147: examinations ?
20) L:161 were applied
21) There are many others, I can't report everything, please try to revise carefully your paper.
22) Could you please add some some vertical line in your tables ? at least to separate between IG and CG, some of them are unclear. Please also put the first titles of tables in Bold
23) At what moment the study was conducted ? I think you should specify it.
24)If it was at the same time as the pandemic crisis, I think you should add a little sentence about that. You can cite this paper who showed the importance of physical activity as a preventive measure against epidemics:
*DOI: 10.1101/2022.01.23.22269214
25) I agree with the last part of your limitations, I know that it's difficult to monopolize and follow seriously a lot of individuals, genereally a small size for non-randomized studies is better for a better monitoring. However, I think that your conclusion is a little bit strong considering the situation, please try to revise it.
I wish the authors good luck for the revision of their paper and for the publication process.
Reviewer 2 Report
Dear authors,
First of all, thank you for your effort in your research. Your research is precious in terms of its subject and content. It is successful to deal with the effects of physical activity in adolescent groups, one of today's biggest problems in different dimensions. After necessary corrections or replies, which I will state below some of my concerns about your research, I think your research is suitable for publication in IJERPH.
abstract
First of all, you stated the positive effects of physical activity on SB behaviors in this section, but it cannot be said that you found this result clearly in some parameters. Please specify the result part more clearly.
Introduction
Your introduction is written in a fluent and literary language, from general to specific, but there are some grammatical errors, please check.
Materials and Methods
-Please make your study design clearer. For example, in what order were all the tests done? Especially before aerobic tests
Have you taken measures to ensure that you have complete rest? Or the order in which you watched the tests. Because you wrote that the research was conducted non-randomly.
-My biggest concern in this section is the number of subjects in your research. But there are many different methods of researching. Therefore, indicate the results of the power analysis that you have done, supporting it with reference and also to determine the number of subjects.
Results
Your tables are very difficult to read in the findings, please arrange the tables as figures. If the tables do not fit, you can shorten the parameters in the tables and give detailed information under the table with the figure legend.
Discussion section
I think that this section is detailed with adequate explanations and resources and does not require any editing. But please check some minor grammatical errors here.
best.
Reviewer 3 Report
Effects of a physical exercise program and health advice on the 2 sedentary behavior of adolescents” focuses on a current theme, revealing characteristics for publication. Features an informative title and the purpose is clear. The methodology is sufficiently described with the application of scientific language. The presentation of the results is clear, the small sample size is highlighted, leaving the challenge for new studies with a greater number of observations. Therefore, the following changes are recommended:
Keywords:
It would be advisable to use keywords that are in the Mesh.
Introduction
I suggest a short paragraph on the meaning of The Municipal Human Development Index (MHDI)
Materials and Methods
I suggest in point 2.2.1 a brief summary of the different economic classes mentioned in the results. The issue of portable technologies possession should also be better explained, namely what is the purpose of this assessment / its influence on the results.
Conclusion
It is important to mention that the conclusion is related to the study sample.
Round 2
Reviewer 1 Report
Dear authors,
Thank you for the efforts you put in this revision, it was a good job.
I would like also to thank you for your clear responses to reviewers and for taking into consideration all my comments.
I wish you good luck for the publication process.
Reviewer 2 Report
Dear Authors,
Thank you for your effort. Your manuscript is ready to publish